# Revalorization of *Posidonia oceanica* Waste for the Thermochemical Production of Biochar

Julia Moltó [1,2], Mercedes G. Montalbán [3], Samuel S. Núñez [1,2,*] and Juana D. Jordá [4]

1    Chemical Engineering Department, University of Alicante, P.O. Box 99, 03080 Alicante, Spain; julia.molto@ua.es
2    University Institute of Chemical Process Engineering, University of Alicante, Carretera de San Vicente del Raspeig, s/n, 03690 Alicante, Spain
3    Chemical Engineering Department, Faculty of Chemistry, Regional Campus of International Excellence "Campus Mare Nostrum", University of Murcia, 30071 Murcia, Spain; mercedes.garcia@um.es
4    Agrochemistry and Biochemistry Department, University of Alicante, P.O. Box 99, 03080 Alicante, Spain; juana.jorda@ua.es
*    Correspondence: samuel.nr@ua.es

**Abstract:** Every year, many tonnes of *Posidonia oceanica* are removed from Mediterranean beaches to maintain the quality and pleasure of use of the beaches. Most of this waste ends up in landfills, entailing removal costs. In this work, the *Posidonia oceanica* material was characterised, and a washing system was developed to obtain biochar. An adequate washing of the starting biomass was shown to play a key role as it led to an over 90% salt content reduction and, therefore, a decrease in conductivity values. The use of biochar as a soil remediator improves soil properties, carbon sequestration, and plant growth. However, not all types of biochars are suitable for this type of application. Therefore, the properties of biochar made from *Posidonia oceanica* at different temperatures (300, 400, and 500 °C) were studied. All the biochars obtained showed to exceed 10% organic carbon, which is the lower limit to be applied to soils, the maximum percentage having been obtained at 300 °C. In addition, all presented pH values (8.02, 10.32, and 10.38 for the temperatures of 300, 400, and 500 °C, respectively) that were similar to those of other effective biochars for the remediation of acid soils.

**Keywords:** *Posidonia oceanica*; biochar; pyrolysis; biomass; waste management

## 1. Introduction

Biochar is a stable carbon-rich by-product synthesized through thermochemical processes, mainly pyrolysis/carbonization of plant and animal-based biomass [1]. Pyrolysis requires an absence of oxidizing agent as well as moderate temperatures (400–600 °C) to obtain a high solid biochar yield [2]. Conversely, high temperatures are used to achieve high yields of liquid bio-oil and syngas [3,4]. The yields of each pyrolysis product (biochar, bio-oil, and syngas) also depend on the heating rate and the vapor residence time in the reactor. The production of biochar is favored by lower heating rates (0.1–10 °C/s) and higher vapor residence time [4]. In recent years, the role of biochar has been studied in the domain of environmental management. It promises to improve agriculture and the environment in several ways: (i) through soil remediation, due to its high organic carbon content, which implies high stability in soils and excellent nutrient-retention properties [5]; (ii) through waste management, because waste biomass is widely used for biochar production due to cost-effectiveness and food security advantages [6]; (iii) via climate change mitigation, because biochar can be carbon-negative and hence be used to remove carbon dioxide from the atmosphere [7]; and (iv), thanks to bioenergy production, based on the use of the gases generated in the pyrolysis processes [1,8]. The elemental composition of biochar and its properties are determined by the biomass material employed and the features of the carbonization process [9]. Biomass waste includes forestry and agricultural residues such

as wood, leaves, trunks, peels or husks, and excreta from humans and animals. In the case of plant-based biomass, there are three widespread lignocellulosic components: cellulose (40–60 wt%), hemicellulose (15–30 wt%), and lignin (10–25 wt%). Their percentages depend on the biomass source [4].

*Posidonia oceanica*, one of the most abundant aquatic plants in the Mediterranean Sea and especially in coastal areas, is characterized by high contents of cellulose (38 wt%), hemicellulose (21 wt%), and lignin (27 wt%) [10]. *Posidonia oceanica* is key to the preservation of coastal ecosystems because it produces large amounts of oxygen and represents a source of food, shelter, and protection for many fish species and other underwater animals [11]. In addition, it prevents coastal erosion by reducing wave energy. *Posidonia oceanica* is composed of roots, stems, and long leaves and forms continuous meadows that are 0.5 m to 40 m deep. When the leaves die, they lose their photosynthetic capacity and are separated from the rest of the plant. In this way, a large amount of leaves accumulate on beaches, where they decompose [12]. This waste reaches $5 \cdot \times 10^6$ to $5 \times 10^7$ tonnes per year in Mediterranean areas [10]. In summer, *Posidonia oceanica* wastes need to be removed from touristic areas because of unpleasant odours, mainly due to the formation of hydrogen sulphide, and the presence of insects, such as beach flies. The process of cleaning beaches entails a significant loss of sand, which translates into a high annual cost. These marine residues are usually disposed of in landfills, at a considerable cost for the local authorities [13].

In recent years, a number of studies have focused on the possible revalorization of the renewable *Posidonia oceanica* waste. It has been used as a source of cellulose to reinforce conventional polymers such as polyethylene [14] and biodegradable packaging materials based on starch [13,15], gluten protein [16] or polylactic acid [17–19]. *Posidonia oceanica* fibres have also been studied as a low-cost and renewable adsorbent for removing dyes, such as methylene blue [20] or basic blue 41 [21] from aqueous solution, and for removing heavy metals [22,23] and organic compounds [24]. Orthophosphate, which contributes to water quality deterioration and eutrophication in lakes, rivers, and coastal areas, was also successfully adsorbed into *Posidonia oceanica* fibres [25]. They have also been shown to be a suitable source of lignocellulosic fibres to produce pulp and paper [26]. In addition, a crystal absorber constituted of *Posidonia oceanica* fibrous spheres can be made to develop acoustic materials [27]. Recently, a study on the benefits of *Posidonia oceanica* leaf extracts to treat diseases revealed anti-inflammatory properties [28] and the inhibition of human cancer cell migration [29].

However, the potential use of *Posidonia oceanica* waste to produce biochar via thermochemical processes has hitherto been scarcely investigated. Chiodo et al. studied the feasibility of producing bio-oil and biochar from *Posidonia oceanica* by pyrolysis. They found stability and alkalinity characteristics of interest to apply in soils [3]. Cataldo et al. synthesized activated biochar from the pyrolysis of dead *Posidonia oceanica* leaves [9]. It was tested as an adsorbent material for toxic metal ions [30]. In this way, the use of thermally treated *Posidonia oceanica* residues has been shown to be a low-cost adsorbent material with high selectivity toward phosphate from real wastewater [30].

It is very important to characterise biochar when it is generated because the content of multivalent metal elements can improve its efficiency. Dai et al. [31] studied the role of a biochar's calcium content in removing/recovering phosphorus. In addition, it has been shown that the $CaCO_3$ present in biochar improves its stability and $CO_2$ sequestration efficiency [32]. Additionally, it has been shown that CaO can favour the formation of $H_2$ and $CH_4$ in pyrolysis processes [33]. Other authors have also studied how particle size affects pH levels, micronutrients, and macronutrients [34]. For all these reasons, when the applicability of biochar– that has been generated from an insufficiently studied biomass–needs to be verified, it is important to conduct an exhaustive characterisation.

This work studies the main characteristics of biochar produced at different temperatures from washed *Posidonia oceanica*. Water consumption should be reduced for washing, because water is too precious of a resource and should not be used to clean waste in such a considerable amount. For this reason, it is proposed that this flushing be carried out with

water from sewage treatment plants. The physicochemical characteristics of the reclaimed water are within the limits permitted for this type of water. The permitted uses include environmental use, to which the present study can be assimilated since the activity consists of washing to dissolve the salinity of the Posidonia oceanica.

The aim was to optimise the production of biochar from this biomass, thus avoiding the displacement of tonnes of *Posidonia oceanica* to landfills every year and giving the waste a second life by transforming it into biochar. The production of biochar from biomass has an average energy cost of 1.5 MJ/kg [35], which, if translated into money, would have a production cost of 0.05 €/kg (Average price per kWh in Spain in the year 2021 of 0.12 €) [36], which seems economical when compared to the average price of biochar already manufactured. However, it would be interesting to carry out an economic study and a life cycle analysis, in which the possible use of the biochar produced could be considered. This way, it would be possible to exploit its numerous advantages, maintain the value of the product for as long as possible, minimise resources, and contribute to the circular economy.

## 2. Materials and Methods

### 2.1. Biomass Preparation

*Posidonia oceanica* leaves were collected from El Campello beach in Alicante (Spain). When the plant was collected from the beach, it was dried in the sun to remove part of its water content and to separate sand leftovers. To guarantee that the samples were completely dried, *Posidonia oceanica* leaves were inserted into an oven at 110 °C for 24 h. Once the samples were dry, they were ground with a Restch SM 200 cutting mill. The resulting particles had a mean diameter of 1 mm.

### 2.2. Characterization of the Posidonia oceanica Samples

The samples were characterized in terms of their elemental analysis, proximate analysis, and calorific value.

Elemental analysis was performed on *Posidonia oceanica* samples using a Thermo Finnigan Flash 1112 Series Elemental Analyzer (Thermo Fisher Scientific, Waltham, MA, USA). The elemental analysis was carried out in accordance with the standard test DIN 51732:2014 [37] based on the European Biochar Certificate. X-ray fluorescence of *Posidonia oceanica* was conducted with a PW2400 automatic sequential X-ray fluorescence spectrometer (Philips Co., Westborough, MA, USA).

Low and high calorific values (LCV and HCV) of *Posidonia oceanica* were determined using an AC-350 calorific bomb (Leco Corporation, St. Joseph, MI, USA) in accordance with the standard test DIN 51900-1:2000 and based on the European Biochar Certificate [38]. In essence, the experiment consisted of measuring the temperature increase in an adiabatic bath after the sample's combustion. This increment was used to calculate the energy released during combustion.

The proximate analysis of *Posidonia oceanica* consisted of determining the moisture content, ash, volatile matter, and fixed carbon. The moisture content of *Posidonia oceanica* wastes was determined in triplicate based on the DIN 51718:2002 standard [39] through drying in a convection oven at 110 °C as previously mentioned. The ashes were determined in triplicate in accordance with the standard test DIN 51719:1997 and based on the European Biochar Certificate [40]. In short, it consists of combusting the samples at 550 °C in a muffle furnace (Hobersal, Barcelona, Spain) to constant weight. Values of moisture content and ashes were also determined by thermogravimetric analysis (TGA) and derivative thermogravimetric analysis (DTG), which are also used to determine the volatile matter and fixed carbon [41]. This method measures the amount and rate of change in the sample mass as a function of temperature. The temperature at the minimum or maximum peak of the DTG curve represents the temperature at which the samples are decomposed. The assays were carried out in a Perkin Elmer STA600 thermogravimetric analyzer.

The *Posidonia oceanica* sample was also characterized in terms of its composition by thermochemical analysis, such as combustion and pyrolysis. As is known, combustion takes place under oxidant conditions with air (21% oxygen, 79% nitrogen), while pyrolysis occurs in an inert atmosphere (nitrogen only). These experiments were performed in the same thermogravimetric analyzer but following a different program: (i) 7 mg of sample were fed to the equipment; (ii) the gas flow of air or nitrogen (100 mL/min) was open for the combustion and pyrolysis experiments, respectively; (iii) the sample was heated from room temperature (30 °C, approximately) to 950 °C at a heating rate of 10 °C/min; (iv) this temperature was held for 5 min; and (v), the sample was finally cooled from 950 to 30 °C at a cooling rate of 90 °C/min.

## 2.3. Washing of the Posidonia oceanica Samples

The samples were intensively washed with a continuous and constant flow of tap water to remove as much salt content as possible. This process is key when using synthesized biochar to apply to the soil. A volumetric flow of 8.2 mL/s of water via a peristaltic pump was used. It operated at 150 rpm. The sample stirring rate was 300 rpm. The process was controlled by measuring the water's ionic conductivity before and after the washing. When the water's ionic conductivity before and after the treatment was approximately equal, the treatment terminated, and the sample was dried at 105 °C for 24 h then ground to 1 mm.

The washed *Posidonia* samples were dried and then analysed by X-ray fluorescence to verify washing efficiency. Determining the reduction of the halogen emitted during combustion allowed a comparison with the non-washed samples. The experiments were conducted following the SW-846 Test Method 5050 of the U.S. Environmental Protection Agency (EPA) [42]. Fluoride, chloride, nitrite, bromide, nitrate, phosphate, and sulphate contents were defined using calorific bomb and ionic chromatography as a detection method. In short, 10 mL of a $NaHCO_3/Na_2CO_3$ (0.03 M/0.024 M) aqueous solution and 0.5 g of biomass were introduced into the calorific bomb for combustion. After combustion, the resultant solution was collected and diluted with distilled water in a 100 mL flask. An aliquot of this solution was analyzed using a Dionex DX-500 ionic chromatograph (Thermo Fisher Scientific Inc., Waltham, MA, USA). Each halogen's concentration was determined in ppm. The concentration of the ion per gram of sample in µg/g, $C_o$, was calculated with the Equation (1):

$$C_o = \frac{C_{comb} \times V_{comb} \times DF}{W_o}. \tag{1}$$

where $C_{comb}$ is the ion concentration in the solution in ppm, $V_{comb}$ is the collected volume in the calorific bomb after combustion (10 mL), $DF$ is the dilution factor (10), and $W_o$. is the sample mass.

## 2.4. Biochar Preparation and Characterization

The biochar was prepared via pyrolysis of dried 1 mm particles of washed *Posidonia oceanica* in a muffle furnace (Hobersal, Barcelona, Spain) at three temperatures: 300, 400, and 500 °C, which were selected based on a previous study by Keiluweit et al. [43]. The tests started with the heating of the muffle using a heat ramp for 45 min. It was then kept at a constant temperature for one hour.

The biochar characterization was performed following the International Biochar Initiative [44], conducting an elemental analysis, an X-ray fluorescence analysis, the measurement of pH, electrical conductivity, percentage of $CaCO_3$, organic carbon, percentage of ash, bulk density and thermogravimetric analysis.

An elemental analysis of the biochar was performed using a Thermo Finnigan Flash 1112 Series Elemental Analyzer (Thermo Fisher Scientific, Waltham, MA, USA). It was carried out in accordance with the standard test DIN 51732:2014 based on the European Biochar Certificate [37]. An X-ray fluorescence biochar analysis was executed in the same way as for the *Posidonia oceanica* leaves in a PW2400 automatic sequential X-ray fluorescence spectrometer (Philips Co., Westborough, MA, USA).

For the pH measurements, a Crison Basic 20 pH meter (Mettler Toledo, Columbus, USA) was used. This parameter was estimated according to the international norm ISO 10390:2005 and following the European Biochar Certificate [45]. This standard describes an instrumental method for routine pH determination using a glass electrode in a 1/5 (*v/v*) suspension of biochar sample in water. To summarize, 5 mL of biochar sample (approximately 2 g of dried sample) were added to 25 mL of distilled water and stirred for 1 h. The pH was then measured.

The electrical conductivity of the biochar samples was measured following the norm DIN ISO 11265:1997-06 as recommended by the European Biochar Certificate [46]. A total of 20 g of biochar sample were added to 200 mL of distilled water and stirred for 1 h. The suspension was then filtered, and the conductivity of the filtered water was measured. The salt content was estimated using an empirical factor based on the conductivity of a 0.01 M KCl solution and expressed as mg KCl/L. Three measurements were taken for each biochar sample.

The amount of total inorganic carbon was mainly due to $CaCO_3$, its determination in the samples was based on the reaction that takes place when biochar is attacked by hydrochloric acid:

$$CaCO_3 \text{ (s)} + 2HCl \text{ (l)} \rightarrow CO_2 \text{ (g)} + CaCl_2 \text{ (s)} + H_2O.$$

The $CaCO_3$ percentage was analyzed following the norm DIN 51726:2004-06 as published in the European Biochar Certificate [47]. A Bernard calcimeter was used, which consisted of attacking the carbonates with acid and quantifying the released $CO_2$.

The organic carbon was calculated by determining the difference between the total carbon content obtained by the biochar's elemental analysis and the inorganic carbon determined from the $CaCO_3$ percentage.

For their part, the ashes were analysed according to the DIN 51719: 1997 standard and the European Biochar Foundation and Certificate [48], as in the case of the *Posidonia oceanica* samples. In essence, the method consisted of the combustion of the samples at 550 °C in a muffle oven (Hobersal, Barcelona, Spain) at a constant weight. The results were obtained by weighing the difference between the initial and final masses.

Furthermore, the ash content values were also determined by thermogravimetric analysis (TGA) and derivative thermogravimetric analysis (DTG), which were equally used to determine the volatile matter and fixed carbon [41]. Once the temperature program was completed, the decomposition curves of the different biochar were obtained. The percentages of ashes contained in the different samples analysed at 550 °C and 815 °C could be determined at this point using the following expression:

$$\% \text{ } Ashes \text{ } (T) = \frac{Sample \text{ } mass \text{ } (T)}{Sample \text{ } mass \text{ } (120 \text{ }°C)} \cdot 100 \tag{2}$$

where sample mass (*T*) is the final sample amount at the experiment temperature and sample mass (120 °C) is the initial mass (on a dry basis).

The bulk density was defined as the amount of sample weight that can be accumulated in each unit volume. It is necessary to consider both the volume occupied by the solid and the air between its particles. The analysis was carried out according to the DIN 51705:2001-06 standard as published in the European Biochar Certificate [49]. The sample was put into a graduated cylinder, in this case, a 50-mL burette, and placed on a balance to estimate its mass. The sample volume was read after having compressed the biochar as much as possible in the burette to prevent empty voids.

### 3. Results and Discussion

*3.1. Characterisation of the Posidonia oceanica Samples*

The characterisation results of *Posidonia oceanica* (ultimate and proximate analysis) show the main physic-chemical properties of the starting biomass (Table 1). *Posidonia oceanica* had

previously been characterised by other authors [3], and a comparison revealed very similar values to those already published. It is important to determine the H/C atom ratio and the O/C atom ratio because O/C ratios have been shown to correlate well with biochar stability [50] and H/C ratios change substantially after thermochemical treatment [43]. Finally, wt% of protein was also calculated. The protein fraction was estimated based on the nitrogen content (wt%) of the biomass using a nitrogen factor (NF) of 6.25 [51].

**Table 1.** Physicochemical characterization of the *Posidonia oceanica* wastes (unwashed).

| Materials | *Posidonia oceanica* |
|---|---|
| Elemental Analysis (wt%) | |
| C | 29.28 |
| H | 3.62 |
| N | 0.77 |
| S | 0.87 |
| H/C atom ratio | 0.12 |
| O/C atom ratio | 0.72 |
| Protein (wt%) [a] | 4.81 |
| LCV (kcal/kg) | 2496.44 |
| HCV (kcal/kg) | 2686.57 |
| Proximate analysis (wt%) | |
| Ash [b] | 44.46 |
| Moisture content | 8.54 |
| Volatile matter | 51.61 |
| Fixed carbon | 12.07 |

[a] wt% of protein = wt% of nitrogen × NF. [b] Muffle oven.

To summarise the characterisation of *Posidonia oceanica*, the results obtained in the different thermogravimetric tests under different oxygenation conditions: combustion and pyrolysis are shown (Figure 1).

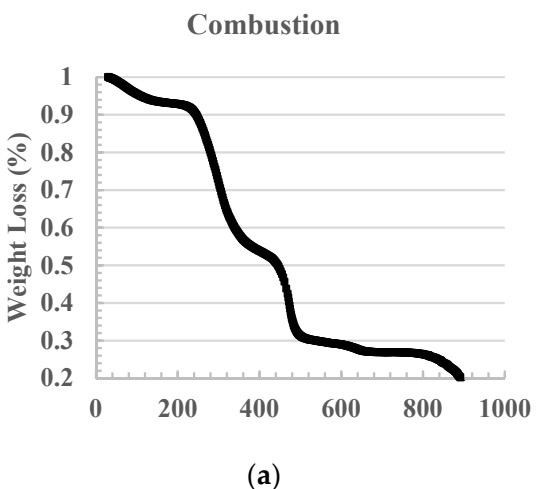

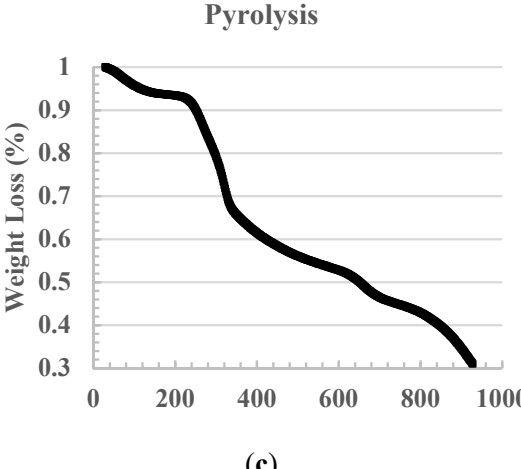

(a)　　　　　　　　　　　　　　　　　　　　　(c)

**Figure 1.** *Cont.*

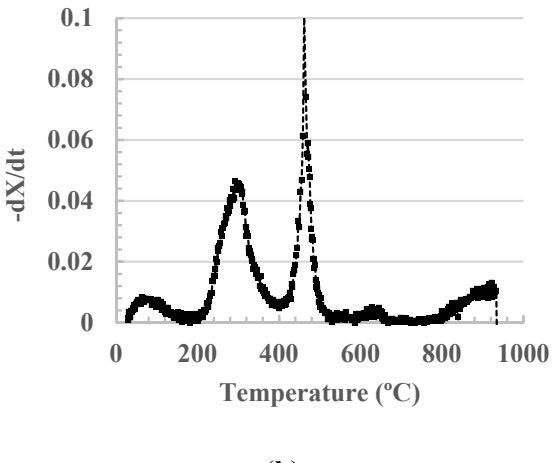

**(b)**

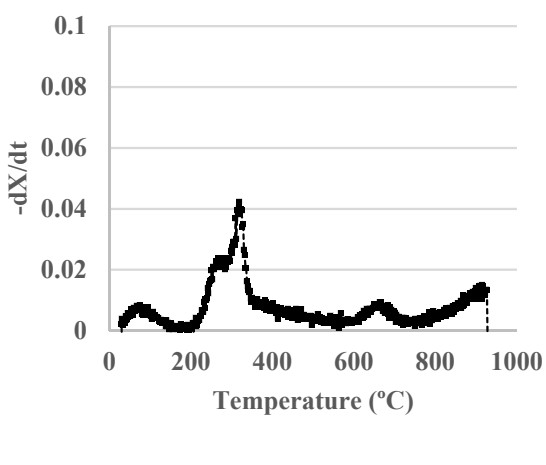

**(d)**

**Figure 1.** (**a**) TG from biomass combustion; (**b**) DTG from biomass combustion; (**c**) TG from biomass pyrolysis; (**d**) DTG from biomass pyrolysis.

Regarding combustion (Figure 1a,b), there was a first stage of weight loss, which corresponded to the sample's moisture loss. From 250 °C onwards, the decomposition rate increased, which may be due to the loss of volatile components in the sample. According to the immediate analysis, the volatile components represented over 50% of the weight of the total dry mass (Table 1). In this phase, the organic macromolecular chains were broken, leading to the formation of smaller molecules. When in contact with oxygen at a temperature above their flash point, volatilised gases rapidly oxidise and burn. Above 500 °C, the resulting mass fraction corresponds to the oxidation of the carbonaceous residue formed or the biochar. Concerning differential thermogravimetry, there were two distinct peaks: one at 288 °C, which corresponds to the decomposition of hemicellulose; and another at 470 °C, which corresponds to the thermal decomposition of cellulose.

The biomass pyrolysis results were presented in the same way as for combustion (Figure 1c,d). Unlike the case of combustion, the decomposition stages were not very abrupt. First, as the temperature increased, a drying process took place and, possibly, the release of some volatile compounds. From 250 °C onwards, the low molecular weight compounds began to volatilise, up to about 350 °C. The remaining mass fraction of the gas was then released. The rest of the mass fraction corresponded to ash and solid carbonaceous residues that were not oxidised by the atmospheric conditions under which the decomposition took place. The differential thermogravimetry revealed two other peaks: one at 273 °C, corresponding to the decomposition of hemicellulose, and the other at 319 °C, corresponding to the thermal decomposition of cellulose. Lignin decomposition was difficult to identify as it decomposes over a wide range of temperatures. The 660 °C peaks may have corresponded to the decomposition of this compound.

### 3.2. Reduction of the Salt Content of Washed Posidonia oceanica

The salt content reduction was determined by X-ray fluorescence. The concentration of Na and Cl was reduced after this treatment by 92% and 95%, respectively (Table 2).

**Table 2.** X-ray fluorescence results before and after *Posidonia oceanica* washing.

| Materials | Unwashed Oceanic Posidonia (wt%) | Washed Oceanic Posidonia (wt%) | Reduction (%) |
|---|---|---|---|
| Na | 5.61 | 0.43 | 92.34 |
| Cl | 14.88 | 0.72 | 95.16 |
| Mg | 2.01 | 1.11 | 44.78 |
| K | 1.41 | 0.15 | 89.36 |
| P | 0.07 | 0.05 | 28.57 |

Different origins of biomass are currently being considered as a substitute for fossil fuels. This is mainly due to the fact that their $CO_2$ emission balance is zero, and they represent renewable fuel. However, combustion produces the volatilization of certain elements that can pose a risk to the environment and people. The most volatile of these elements is chlorine.

The aim of the wash was mainly to reduce the quantity of salts contained in the *Posidonia oceanica*. The main form of chlorine release is hydrochloric acid, which entails an additional risk due to dioxins. Highly toxic chemical compounds generated during combustion processes in the presence of aromatic chemical compounds and chlorine could be produced. Therefore, the amount of chlorine should be reduced as much as possible. The concentration ($C_o$) values for each halogen emitted during the samples' combustion before and after the washing were determined (Table 3).

**Table 3.** Values of $C_o$ for each halogen emitted during combustion.

| | $C_o$ (µg/g) | | | | | | |
|---|---|---|---|---|---|---|---|
| | $F^-$ | $Cl^-$ | $NO^{2-}$ | $NO^{3-}$ | $Br^-$ | $PO_4^{3-}$ | $SO_4^{2-}$ |
| Before washing | 26.8 | 34,461.3 | 458.5 | 7020.4 | 204.7 | 25.1 | 12,170.2 |
| After washing | 26.6 | 1863.7 | 1152.3 | 13,975.6 | 185.4 | 23.2 | 7468.6 |

For their part, sulphate concentrations in the sample were also reduced after washing. It is worth noting that sulphur forms a wide range of both organic and inorganic compounds in biomass and in different oxidation states. When it appears as sulphate, it is taken by plant roots up to the leaves, where a reduction process takes place [52].

Nitrates and nitrites showed an increase with respect to the unwashed plant. This may be due to the fact that *Posidonia oceanica* absorbed the nitrates and nitrites present in the washed water used (tap water). Indeed, previous analyses have shown the existing content of nitrates and nitrites in Spanish public drinking water [53]. In nature, plants use nitrates as an essential nutrient. From a commercial perspective, most nitrates are used in inorganic fertilisers. Nitrates and nitrites are also used in food preservation, in some medicines, and in the manufacture of ammunition and explosives [52].

Fluoride, bromide, and phosphate showed little variation, although they can be considered almost nil due to low concentrations in all samples.

*3.3. Biochar Characterization*

The biochar characterization was carried out following the International Biochar Initiative Product Testing Guidelines for Biochar That is Used in Soil [44]. The results of the different analyses summarised (Table 4). As is observable, the values obtained for all elements decreased as the temperature increased. Also worthy of note is the H/C ratio: the higher a fuel's H/C ratio, the higher its $CO_2$ emissions when compared to those of water, as its carbon content will be lower than its hydrogen content. Therefore, it is possible to conclude that the higher the temperature of biochar production, the lower its $CO_2$ emissions.

The International Biochar Initiative classifies biochar into three categories depending on its organic carbon content, setting a lower limit that must be exceeded for any biochar to be used. The lower limit is 10% organic carbon. A biochar with more than 60% would belong to class 1, with more than 30% it would belong to class 2, and below 30%, it would belong to class 3. The analysis allows concluding that the biochar produced at 300 °C belongs to class 2. In contrast, those produced at higher temperatures belong to class 3.

The comparison between biochar at 400 °C and biochar at 500 °C shows that the carbon content continues to decline, but the ash content decreases (Table 4). This is probably due to the formation and emission of polycyclic aromatic compounds (PAHs), as these tend to form under conditions of incomplete organic matter combustion at temperatures above 400 °C.

Regarding the results obtained for pH, conductivity, salt content, and bulk density, no large variations were found for the parameters studied. One exception was pH, which

varied by two units between two samples (biochar 300 °C and biochar 400 °C). These parameters were compared to those in the literature for biochar made from plant residues and were found to be within the normal range for application to soils. In fact, Qayyum et al. [54] performed a germination test on a plant waste biochar with very similar characteristics to that found for *Posidonia oceanica* biochar. The author obtained germination results of more than 85%. Biochar obtained from *Posidonia oceanica* can therefore be assumed to be suitable for soil remediation. In addition, since biochar has relatively high conductivity and pH, it can be considered as feasible to apply it as a remediation for acidic soils. Such feasibility has, in fact, already been an object of study, and positive results were obtained [55].

**Table 4.** Characteristics of the different biochars obtained from washed *Posidonia oceanica* leaves.

| | Biochar 300 °C | Biochar 400 °C | Biochar 500 °C |
|---|---|---|---|
| Nitrogen (wt%) | 0.93 | 0.61 | 0.56 |
| Carbon (wt%) | 41.07 | 28.31 | 26.60 |
| C (inorganic) (wt%) | 2.74 | 3.52 | 4.21 |
| C (organic) (wt%) | 38.32 | 24.79 | 22.40 |
| Hydrogen (wt%) | 2.50 | 1.17 | 0.65 |
| H/C organic | 0.07 | 0.05 | 0.03 |
| Ashes (wt%) | 44.46 | 66.42 | 57.02 |
| Oxygen (by difference) | 11.04 | 3.50 | 15.16 |
| $CaCO_3$ (wt%) | 22.86 | 29.36 | 35.06 |
| Conductivity (dS/m) | 3.19 | 3.31 | 3.08 |
| Salt content (mg KCl/L) | 1682.56 | 1745.92 | 1624.83 |
| pH | 8.02 | 10.32 | 10.38 |
| Bulk density (kg/m$^3$) | 258.31 | 256.26 | 245.80 |
| % yield of biochar | 75 | 67 | 57 |
| Fluorescence analysis-semiquantitative analysis | | | |
| Na (wt%) | 0.65 | 0.77 | 0.96 |
| Mg (wt%) | 1.61 | 2.01 | 2.49 |
| Ca (wt%) | 21.6 | 21.9 | 25.5 |
| K (wt%) | 0.25 | 0.24 | 0.27 |
| Fe (wt%) | 1.48 | 1.67 | 1.57 |
| Sr (wt%) | 0.18 | 0.2 | 0.19 |
| Si (wt%) | 2.36 | 2.18 | 2.84 |
| P (wt%) | 0.08 | 0.09 | 0.11 |
| S (wt%) | 1.30 | 1.40 | 1.50 |
| Ni (wt%) | 0.01 | 0.01 | 0.01 |

Finally, the results obtained from the thermogravimetric analysis of the different biochars produced at 300, 400, and 500 °C are presented (Figure 2). As the biochar comes from a plant (*Posidonia oceanica*) pyrolysis, the decomposition of three main components was expected: hemicellulose, which has a decomposition temperature between 220 and 315 °C; cellulose, whose decomposition occurs from 315 °C and up to about 400 °C; and, finally, lignin, whose decomposition is somewhat more difficult, in some cases requiring maximum temperatures of 900 °C to decompose [56]. As presumed, three peaks of matter decomposition were observed in the first biochar (300 °C) and in the third (500 °C). These peaks occurred at the expected temperature ranges, corresponding to the decomposition of hemicellulose, cellulose, and lignin. However, only two peaks were observed in the results obtained for the biochar produced at 400 °C (Figure 2c,d). They corresponded to cellulose (435 °C) and lignin (663 °C) decomposition. It was not possible to observe the cellulose decomposition as it is likely to have taken place during the production of the biochar.

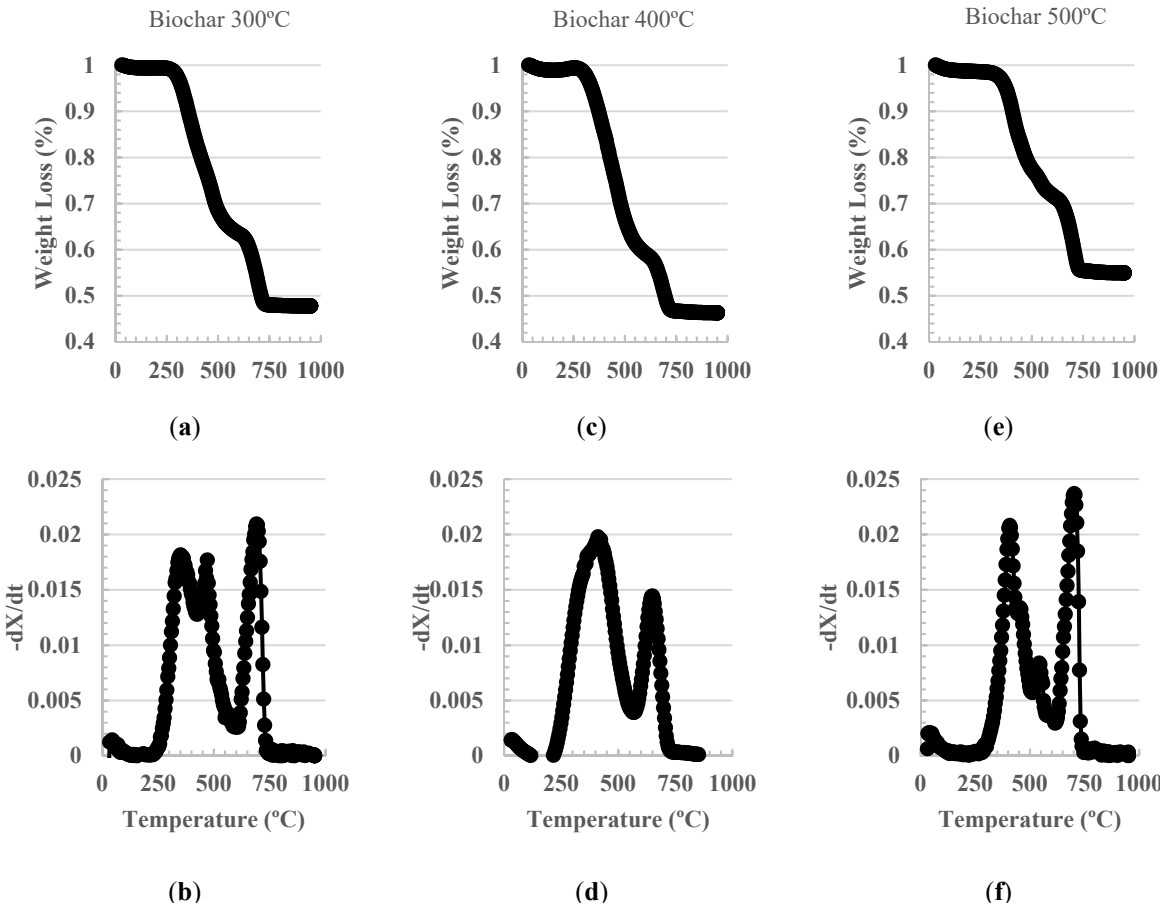

**Figure 2.** (**a**) TG biochar 300 °C; (**b**) DTG biochar 300 °C; (**c**) TG biochar 400 °C; (**d**) DTG biochar 400 °C; (**e**) TG biochar 500 °C; (**f**) DTG biochar 500 °C.

## 4. Conclusions

The aim of this work was to study the possibility of using marine waste to manufacture biochar, which is a problem for tourism because it is deposited on the beaches and decomposes, generating a bad smell and attracting insects. Therefore, it is currently collected and sent to a landfill. Another objective was to examine the feasibility of a soil remediation application. Correctly washing the biomass (*Posidonia oceanica*) before producing the biochar was determined as an important step. In this case, the sea waste had a high salt content and, therefore, a high conductivity, hindering its use in soil remediation. The salt content was reduced by more than 90% after washing.

The biochar was generated at three different temperatures (300, 400, and 500 °C). The H/C ratio was monitored in all of them. In this study, this ratio decreased as the temperature increased, indicating that structural transformations induced a carbonization process. The organic carbon content of all biochars was also determined, and all presented values above 10%, which is the lower limit for application to soil. In addition, the biochar characteristics were compared and found to be like other types of biochars whose toxicity were demonstrated to be null through germination tests. This latter finding also supports the use of this type of biochar for soil remediation, which would thus also allow reducing the amount of waste in landfills.

In conclusion, this study aims to prevent tonnes of biomass waste from going to landfill every year by finding a use for this biomass, for which it was washed, and its physicochemical properties were studied with and without pyrolysis, with the aim of moving from a linear economy to a circular economy, which makes use of 100% of the resources.

**Author Contributions:** Conceptualization, J.M., M.G.M. and J.D.J.; methodology, J.D.J.; formal analysis, S.S.N.; investigation, M.G.M.; data curation, S.S.N.; writing—original draft preparation, M.G.M.; writing—review and editing, S.S.N.; supervision, J.M. and J.D.J.; project administration, J.M.; funding acquisition, J.M. All authors have read and agreed to the published version of the manuscript.

**Funding:** The present study was funded by the Ministry of Science and Innovation (Spain) [grant number PID2019-108632RB-I00] and by Prometheus Programme (Spain) [grant number CIPROM/2021/027].

**Institutional Review Board Statement:** Not applicable.

**Informed Consent Statement:** Not applicable.

**Conflicts of Interest:** The authors declare no conflict of interest.

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
