# Peer review of "Revalorization of Posidonia oceanica Waste for the Thermochemical Production of Biochar"

_applsci, doi:10.3390/app12157422_

Round 1
Reviewer 1 Report
The paper by Molto et al. has a clear important aim: upcycling a waste to produce a useful valuable product, effective for improving soil properties and carbon sequestration.
What it is really missing in the paper is an economic evaluation of the biochar production; such an assessment should be indispensable from the perspective of the circular economy. Can the authors calculate the price per kg of the final product? how much does washing affect the final price? and the pyrolisis?
Water consumption should be reduce to the minimum, or a different method should be looked for to remove salts, because water is a too precious resource and should not be used to clean a waste in such a huge amount. What happens to the waste water produced by the washing? I am sure it becomes another waste to dispose of somehow.
A comparison between the removal costs and the biochar production would give an idea of the feasibility of the process.
At least a mention of these issues should be included in the article.
Check line 240 for the reference
Check lines 246-256: something is missing
Reviewer 2 Report
Paper is acceptable for publishing with some revision. The comments are below.
1. Line no. 28, remove “-“ after plant.
2. Line no. 29, Temperature should be given in range.
3. Line no. 42, give some more reference of above-mentioned line (36-42).
4. Tables and figures should not be cited properly. You introduce the material first, then quote any figures or tables in brackets.
5. Figure 1. Is not clear. You can break it up in the following ways like a, b, c, d and cite it in your MS. Without division it creates confusion to readers. Because you only mentioned figure 1, understanding is complicated.
6. In table 1, 2, 3, and 4 formatting issue. You used ‘,’ (comma) instead of ‘.’ (dot). So, correct it.
7. Recalculate the reduction% in table 2 and also remove % sign from the last line of table 2.
8. the same issue as figure 1. Please separate the figures using the a, b, c, and d formats and mention them appropriately in MS, for example, fig. 2a, fig. 2b.
9. Try to go into greater detail on waste management in the conclusion section. Actual findings and the significance of your investigation.
Other comments
1) In abstract line 14 what does the author mean by saying starting posidonia oceanica material
2) In introduction line 28 correct the format
3) In introduction line 72 give some reference for organic pollutants
4) In introduction check line 83-84 reference
5) Check table 1 whether it should be comma or decimal point in between numbers
